# Data Analysis of Electrical Impedance Spectroscopy-Based Biosensors Using Artificial Neural Networks for Resource Constrained Devices

**Marco Grossi *** and **Martin Omaña**

Department of Electrical Energy and Information Engineering "Guglielmo Marconi" (DEI), University of Bologna, 40136 Bologna, Italy; martin.omana@unibo.it
* Correspondence: marco.grossi8@unibo.it; Tel.: +39-0512093038

**Abstract**

Portable and wearable sensors have gained attention in recent years to perform measurements in many different applications. Sensors based on Electrical Impedance Spectroscopy (EIS) are particularly promising, because they can make accurate measurements with minimum perturbation to the sample under test. Electrochemical biosensors are devices that use electrochemical techniques to measure a target analyte. In the case of electrochemical biosensors based on EIS, the measured impedance spectrum is fitted to that of an equivalent electrical circuit, whose component values are then used to estimate the concentration of the target analyte. Fitting EIS data is usually carried out by sophisticated algorithms running on a PC. In this paper, we have evaluated the feasibility to perform EIS data fitting using simple Artificial Neural Networks (ANNs) that can be run on resource constrained microcontrollers, which are typically used for portable and wearable sensors. We considered a typical case of an impedance spectrum in the range 0.1 Hz–10 kHz, modeled by using the simplified Randles equivalent circuit. Our analyses have shown that simple ANNs can be a low power alternative to perform EIS data fitting on low-cost microcontrollers with a memory occupation in the order of kilo bytes and a measurement accuracy between 1% and 3%.

**Keywords:** biosensors; artificial neural networks; electrical impedance spectroscopy; microcontrollers; Randles circuit; equivalent circuit; memory usage

## 1. Introduction

The market of portable and wearable sensor systems experienced a strong growth in recent years with applications in different fields, such as environmental monitoring [1–4], food quality analysis [5–8], structural health monitoring [9–12], telemedicine [13–16], and microbial detection [17–20]. Among various types of sensing principles, sensors based on Electrical Impedance Spectroscopy (EIS) [21] are very attractive, since EIS can accurately estimate the electrical parameters over a wide range of frequencies by the application of low-amplitude test signals, thus minimizing the perturbation of the sample under test. For example, in 2022, Grossi et al. presented a portable sensor system for the determination of olive oil quality grade and free acidity [22]. The proposed system can be powered by batteries for in-the-field analysis, and the working principle is based on the electrical characterization of an emulsion between a hydro-alcoholic reagent and the olive oil sample. In 2021, Akhter et al. discussed a low-cost and low-power sensing system for real-time

water quality monitoring [23]. Using sensor electrodes made of multi-walled carbon nanotubes with a polydimethylsiloxane substrate, the developed sensor can detect a wide range of nitrate concentrations from 0.01 ppm to 30 ppm. In 2023, Buscaglia et al. presented Simple-Z, a low-cost portable impedance analyzer that can be produced at a cost of only 100 USD [24]. The proposed system is based on the integrated circuit AD5933 by Analog Devices (Wilmington, MA, USA) and has been successfully validated by measuring different $Na_2SO_4$ concentrations in aqueous solution. EIS was also exploited to design biosensors for different analytes, where a bioreceptor is immobilized on the working electrode for selective detection of the target compound [25].

In EIS-based sensors, the electrodes are applied to the sample under test, and the complex impedance (real and imaginary components of the impedance) is measured by the application of a sine-wave voltage signal (peak-to-peak voltage from 1 mV to a few tens of mV) over a wide range of frequencies (from milli-hertz to several hundreds of kilo-hertz). The measured impedance spectrum is then fitted to an equivalent electrical circuit that models the sensor's electrical properties, and the circuit parameters are used to estimate the sample parameters of interest [26,27]. The impedance spectrum fitting to the selected equivalent electrical circuit is normally carried out by specialized software running on a PC and leveraging complex mathematical algorithms [28,29]. Examples of such EIS data fitting software are ZView from Scribner (Southern Pines, NC, USA) [30], Zahner Analysis by Zahner-Elektrik (Kronach, Germany) [31], ZSimpWin by AMETEK Scientific Instruments (Oak Ridge, TN, USA) [32], as well as the free online data fitting tool EIS Studio [33].

In the case of portable and wearable sensor systems, resource-constrained devices (i.e., low-power microcontrollers) are often used as computing devices to minimize energy consumption and extend the battery lifetime. These devices are characterized by limited computational power and memory size, thus making the implementation of EIS data fitting algorithms difficult. As an example, Table 1 reports some details of low-power microcontrollers commonly used in portable sensor systems. In particular, the table reports, for each microcontroller, the type of CPU, the Flash memory size, the Static Random Access Memory (SRAM) size, the characteristics of the integrated analog-to-digital converter (ADC) and digital-to-analog converter (DAC), and the type of integrated wireless communication protocols (if any) [34–40]. As can be seen, the SRAM size of such a device is limited, with values in the range from 256 kB to 512 kB for the microcontrollers that integrate wireless connectivity to be used in the Internet of Things. In the case of microcontrollers integrated in portable or wearable sensor systems without wireless connectivity, the available SRAM size is even lower, with values between 2 kB and 64 kB.

**Table 1.** Examples of low-power microcontrollers.

| Device | CPU | Flash | SRAM | ADC | DAC | Connectivity | Ref. |
|---|---|---|---|---|---|---|---|
| ESP32 | 32-bit LX6 CPU | 448 kB | 520 kB | 2 × 12-bit ADC | 2 × 8-bit DAC | Wi-fi, Bluetooth | [34] |
| STM32WB5MMG | 32-bit Cortex M4 | 1 MB | 256 kB | 12-bit ADC | NA | BLE, Zigbee, OT | [35] |
| CY8C5888LTI-LP097 | 32-bit Cortex M3 | 256 kB | 64 kB | 2 × 12-bit ADC | 4 × 8-bit DAC | NA | [36] |
| MSP430FG6425 | 16-bit RISC CPU | 64 kB | 10 kB | 16-bit ADC | 2 × 12-bit DAC | NA | [37] |
| PIC18F2455 | 8-bit RISC CPU | 24 kB | 2 kB | 10-bit ADC | NA | NA | [38] |
| STM32L073RZT6 | 32-bit Cortex M0+ | 192 kB | 20 kB | 12-bit ADC | 12-bit DAC | NA | [39] |
| ATmega328P | 8-bit RISC CPU | 32 kB | 2 kB | 10-bit ADC | NA | NA | [40] |

In recent years, Artificial Neural Networks (ANNs) have been widely adopted for data analysis in many applications, such as medical image analysis [41,42], multi-sensor fusion data analysis [43,44], and sound classification [45,46]. Studies on the use of ANNs for the estimation of equivalent electrical circuit parameters in EIS applications have also been

reported in the literature. In 2023, Buchicchio et al. presented a study on the use of machine learning models for the estimation of battery state-of-charge from measured EIS data [47]. The use of ANNs to estimate the state-of-charge of lithium-ion (Li-ion) batteries from EIS data has also been investigated by Luo in 2021 [48]. In 2025, Saran et al. compared different machine learning strategies for the interpretation of EIS data to detect the degradation mechanisms in chemical conversion coatings of magnesium alloys [49]. In 2020, Wang et al. proposed an ANN for EIS equivalent circuit parameters estimation to evaluate the corrosion of Q235A steel [50]. In 2025, Zhang et al. presented a convolutional neural network (CNN) to estimate the state-of-charge of lithium-ion batteries from measured EIS data [51]. In 2023, Doonyapisut et al. discussed the use of deep neural networks for the analysis of EIS data and the parameter estimation for different types of equivalent electrical circuits [52]. In 2023, Zulueta et al. investigated the use of deep ANNs for EIS equivalent circuit parameter identification on a lead–acid battery dataset [53].

While the results in [47–53] demonstrated the feasibility of EIS equivalent circuit parameters estimation using ANNs, the employed ANN structure is generally complex, and this results in a large number of network trainable parameters that lead to high memory occupation. In this work, we explore the feasibility of EIS equivalent circuit parameters estimation by considering simple structures for the employed ANN, and we evaluate the accuracy of EIS parameters estimation as function of memory occupation. We considered, as a case study, the equivalent electrical circuit normally used to model electrochemical biosensors, and we tested ANNs with a maximum of two hidden layers. In Section 2, the considered equivalent electrical circuit used to model electrochemical biosensors is presented. In Section 3, the datasets used in the study, the structure of the investigated ANNs, and the metrics used to evaluate the estimation accuracy are presented. In Section 4, the results achieved with the investigated ANNs are discussed and the estimation accuracy as well as the memory occupation are presented. Finally, conclusive remarks are presented in Section 5.

## 2. Equivalent Circuit for Electrochemical Biosensors

Electrochemical biosensors are devices that exploit electrochemical techniques (such as Electrical Impedance Spectroscopy, Square Wave Voltammetry, Differential Pulse Voltammetry, Cyclic Voltammetry, Chrono-Amperometry) for the detection of different types of analytes [54]. In the case of EIS-based electrochemical biosensors, the most common measurement setup is based on a three electrodes system: the working electrode (WE), modified with an immobilized bioreceptor (antibody, nucleic acid, and lectin) for selective detection of the target analyte; the reference electrode (RE), usually made of Ag/AgCl, that provides a stable reference voltage; the counter electrode (CE) that is the source of the current [55–57]. The structure of the WE is presented in Figure 1a, where the biosensing layer represents an interface between the WE and the bioreceptor, and the binding between the bioreceptor and the target analyte changes the biosensor impedance, thus allowing for the measurement of the analyte concentration.

The biosensor impedance spectrum is measured by the application of a sine-wave voltage test signal (over a wide range of frequencies) between WE and RE, while the developed current is measured at the CE. The impedance spectrum is usually fitted to the Randles circuit of Figure 1b, where $R_s$ accounts for the reagent electrical conductivity; $R_{ct}$ is the charge transfer resistance, that is affected by the binding of the target analytes to the bioreceptors; $Z_{CPE}$ is the impedance of a constant phase element that models the non-ideal capacitive behavior of the biosensor at the interface between the WE and the reagent; $Z_W$ is the Warburg impedance that models the ions diffusion process at very low frequencies.

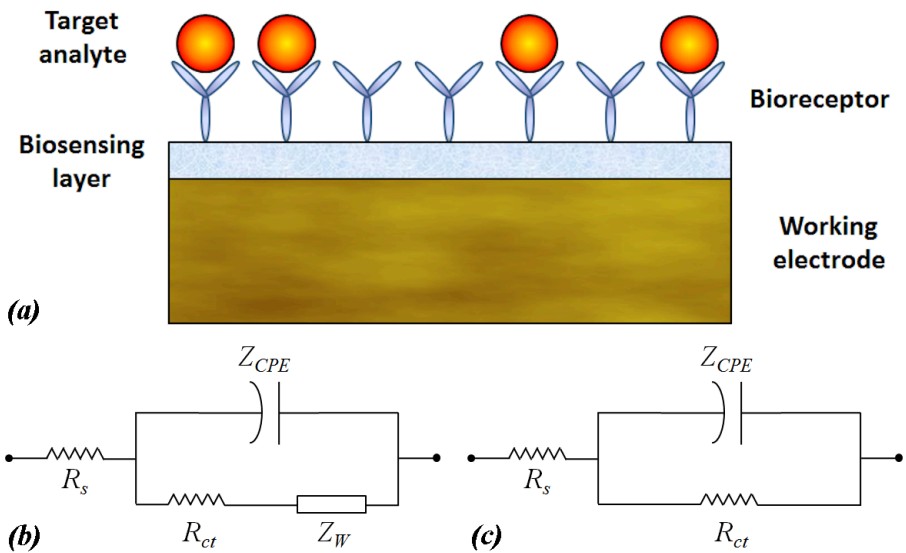

**Figure 1.** Schematic of the working electrode for an electrochemical biosensor (**a**); Randles circuit (**b**); simplified Randles circuit (**c**).

When the frequency of the test signal is not too low, the Randles circuit of Figure 1b can be modified by removing the Warburg impedance $Z_W$, resulting in the simplified Randles circuit of Figure 1c. Given that the impedance of the constant phase element (CPE) can be expressed as follows:

$$Z_{CPE} = \frac{1}{Q(j2\pi f)^\alpha} = \frac{1}{Q(2\pi f)^\alpha} \times e^{-j\frac{\pi}{2}\alpha} = \frac{1}{Q(2\pi f)^\alpha} \times \left[\cos\left(\frac{\pi}{2}\alpha\right) - j\sin\left(\frac{\pi}{2}\alpha\right)\right] \quad (1)$$

where $f$ is the frequency of the sine-wave test signal, $Q$ is the capacitance of the CPE, and $\alpha$ is an empirical parameter (in the range from 0 to 1) that models the non-ideal electrode–reagent interface ($\alpha = 1$ is the case of an ideal capacitor). The impedance of the simplified Randles circuit of Figure 1c (that can be defined by four parameters $R_s$, $R_{ct}$, $Q$, and $\alpha$) can thus be expressed as follows:

$$Z = R_s + \left(\frac{1}{R_{ct}} + \frac{1}{Z_{CPE}}\right)^{-1} = R_s + \left(\frac{1}{R_{ct}} + Q(2\pi f)^\alpha e^{j\frac{\pi}{2}\alpha}\right)^{-1} =$$
$$= R_s + \frac{R_{ct}}{1 + R_{ct}Q(2\pi f)^\alpha \cos\left(\frac{\pi}{2}\alpha\right) + jR_{ct}Q(2\pi f)^\alpha \sin\left(\frac{\pi}{2}\alpha\right)} \quad (2)$$

Some works from the literature that discuss electrochemical biosensors modeled with the simplified Randles circuit are presented in Table 2 [58–65]. For each work, the following information is provided: the target analyte that is detected by the biosensor, the detection range, and the range of variation for the parameters $R_{ct}$ ($\Delta R_{ct}$), $Q$ ($\Delta Q$), and $\alpha$ ($\Delta\alpha$). The range of variation for the parameter $R_s$ is not reported in any work; however, its value should not be higher than a few hundred Ohms, since the supporting reagent is often characterized by high electrical conductivity. As can be seen, changeation range for the model parameters changes significantly for the different biosensors, as it is affected by different elements, such as the material, size, and geometry of the electrodes, the type and concentration of the immobilized bioreceptor, as well as the electrical properties of the reagent.

**Table 2.** Electrochemical biosensors from the literature that are modeled with the simplified Randles circuit.

| Target Analyte | Detection Range | $\Delta R_{ct}$ | $\Delta Q$ | $\Delta \alpha$ | Ref. |
|---|---|---|---|---|---|
| Prussian blue | 0–8 μM | 4.15–14.9 MΩ | 0.82–1.8 μF | NA | [58] |
| KCN | 0–8 μM | 9–13 MΩ | 0.8–4.8 μF | NA | [58] |
| $As_2O_3$ | 0–8 μM | 1.96–4.95 MΩ | 0.8–0.89 μF | NA | [58] |
| E. coli O157:H7 | $10^3$–$10^7$ cfu/mL | 1–15 kΩ | NA | NA | [59] |
| DNA | $10^{-13}$–$10^{-7}$ M | 20–130 kΩ | NA | NA | [60] |
| Bacteria | $10^3$–$10^6$ cfu/mL | 100 Ω–2.5 kΩ | NA | NA | [61] |
| Dengue virus | NA | 10–50 kΩ | 1–4 μF | 0.8–0.9 | [62] |
| Bacteria | $10^4$–$10^8$ cfu/mL | 70–500 Ω | NA | NA | [63] |
| Glucose | NA | 100–600 kΩ | NA | NA | [64] |
| ATP | $15 \cdot 10^{-9}$–$4 \cdot 10^{-3}$ M | 3–30 kΩ | NA | NA | [65] |

## 3. Artificial Neural Networks for EIS Data Fitting

In this section, the proposed strategy to estimate the parameters of the simplified Randles circuit is discussed. We have considered the estimation of the parameters $R_{ct}$ and $Q$ only, since these are parameters normally used to estimate the concentration of the target analyte. In Section 3.1, the datasets used to evaluate the accuracy of the proposed strategy are presented. In Section 3.2, the structure of the investigated ANNs is presented. In Section 3.3, the metrics used to evaluate the estimation accuracy, and the memory occupation are presented. In Section 3.4, the reference value of the accuracy metrics is calculated in the case that the parameter estimation is carried out with a PC-based circuit fitting software.

### 3.1. Considered EIS Datasets

The performance of the ANNs investigated has first been evaluated by different EIS datasets built using an ad hoc developed software program written in LabVIEW v. 14.0 (National Instrument, Austin, TX, USA), and then validated by using a real EIS dataset from the literature [62].

The software-generated datasets were produced by calculating the impedance spectrum (both real component ReZ and imaginary component ImZ) from Equation (2) using the values of the model parameters ($R_s$, $R_{ct}$, $Q$, $\alpha$) and the test signal frequency as inputs. A random noise (of the 0.1%) was added to the calculated impedance value to simulate the uncertainty in the impedance measurement.

The software-generated datasets were produced by considering values of the simplified Randles circuit parameters randomly distributed in the following ranges: from 100 Ω to 400 Ω for $R_s$, from 5 kΩ to 60 kΩ for $R_{ct}$, from 0.5 μF to 5 μF for $Q$, and from 0.75 to 0.95 for $\alpha$. The selected range for the test signal frequency was from 0.1 Hz to 10 kHz.

Three different software-generated datasets, each with 1000 samples, were produced. Dataset A was built by using five test frequencies logarithmically distributed in the range 0.1 Hz–10 kHz. Dataset B was built by using 15 test frequencies logarithmically distributed in the range 0.1 Hz–10 kHz. Dataset C was built by using 25 test frequencies logarithmically distributed in the range 0.1 Hz–10 kHz. Thus, the number of inputs for each sample is 10 for Dataset A, 30 for Dataset B, and 50 for Dataset C (the values of ReZ and ImZ for each test frequency), while the number of outputs for each sample is two for all datasets (the nominal values of $R_{ct}$ and $Q$).

The real EIS dataset used to validate the proposed ANN structures has been presented by Oliveira et al. [62], where an EIS-based biosensor was built by the immobilization of concanavalin A on gold microelectrodes to detect dengue virus in biological samples. The dataset is composed of 18 samples, with 6 samples that are positive to dengue fever (DF), 6 samples that are positive to dengue hemorrhagic fever (DHF), and 6 samples that are

negative to dengue virus (DN, i.e., dengue negative). The values of the parameters $R_{ct}$, $Q$, and $\alpha$ for the 18 samples of the dataset are presented in Table 3, while a value of 200$\Omega$ was assumed for the parameter $R_s$ in all samples.

**Table 3.** Dataset from the experimental measurements of an EIS-based biosensor for the detection of dengue virus [62].

| Sample Type | $R_{ct}$ (k$\Omega$) | $Q$ (µ) | $\alpha$ |
|:---:|:---:|:---:|:---:|
| DF | 33.80 | 3.94 | 0.79 |
| DF | 37.20 | 4.34 | 0.79 |
| DF | 38.90 | 4.54 | 0.78 |
| DF | 33.79 | 2.99 | 0.80 |
| DF | 38.34 | 3.42 | 0.80 |
| DF | 43.73 | 3.59 | 0.79 |
| HDF | 32.00 | 2.72 | 0.85 |
| HDF | 29.10 | 2.47 | 0.86 |
| HDF | 27.86 | 2.36 | 0.86 |
| HDF | 42.31 | 2.67 | 0.86 |
| HDF | 37.01 | 2.35 | 0.86 |
| HDF | 35.20 | 2.17 | 0.86 |
| DN | 19.22 | 1.63 | 0.88 |
| DN | 21.11 | 1.48 | 0.88 |
| DN | 20.19 | 1.55 | 0.88 |
| DN | 19.51 | 1.67 | 0.88 |
| DN | 21.15 | 1.68 | 0.88 |
| DN | 20.34 | 1.58 | 0.88 |

*3.2. Neural Network Structures*

For each tested dataset, 11 different fully connected ANN structures were investigated: 6 ANN structures are characterized by a single hidden layer with number of neurons of 8, 16, 24, 32, 48, and 64, respectively, while 5 ANN structures are characterized by two hidden layers with number of neurons for hidden layer 1 and hidden layer 2 of (16, 8), (24, 12), (32, 16), (48, 24), and (64, 32), respectively. The maximum number of hidden layers was 2 to maintain the number of trainable parameters (i.e., weights and bias), and thus the memory occupation, to an acceptable level. The number of neurons for the input layer is set as the number of input variables, that is 10 for Dataset A, 30 for Dataset B, and 50 for Dataset C. The number of neurons for the output layer is set to two for all ANN structures and all datasets, representing the two parameters of the simplified Randles circuit to be estimated ($R_{ct}$ and $Q$). The used activation function is Rectified Linear Unit (ReLu) for the neurons in the hidden layers and Linear for the neurons in the output layer. The ANN structure for the case of Dataset A, and a single hidden layer with eight neurons is presented in Figure 2. All the ANN structures were implemented using the Keras framework in Anaconda Python distribution v. 2.6.0. The 1000 samples of the generated dataset were divided into 800 samples for the ANN training (20% used for validation) and 200 samples for the ANN testing. The network training was carried out using the Adam optimizer and mean squared error as a loss function, on 200 epochs with a batch size of 32.

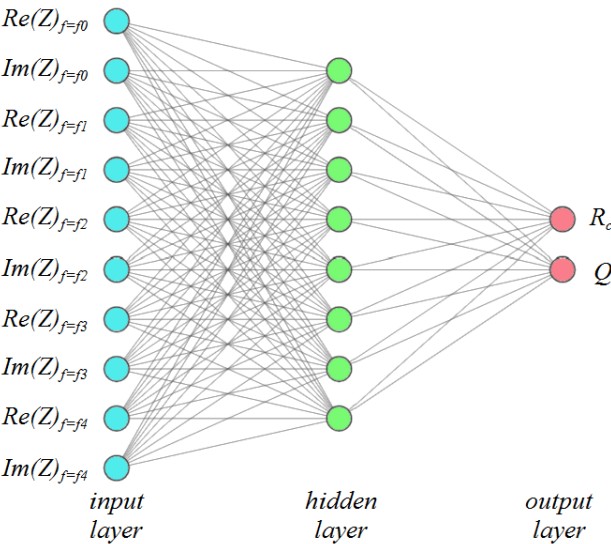

**Figure 2.** Schematic of the ANN structure for the case of Dataset A, and a single hidden layer with eight neurons.

### 3.3. Performance Metrics

A set of metrics were defined to evaluate the performance of the investigated ANNs to accurately estimate the simplified Randles circuit parameters ($R_{ct}$ and $Q$) as well as to evaluate the cost in terms of memory occupation for the ANN trainable parameters.

The accuracy in the equivalent circuit parameters estimation was evaluated, for both $R_{ct}$ and $Q$, by two different metrics: the relative error of the estimated parameter and the normalized mean squared error.

The relative error of the estimated parameter $P$ (with $P = R_{ct}$ or $Q$), can be expressed (in percent) as follows:

$$\Delta E_P = 100 \cdot \frac{|P_{estimated} - P_{nominal}|}{P_{nominal}} \tag{3}$$

where $P_{estimated}$ is the estimated value of the parameter $P$, while $P_{nominal}$ is its nominal value. The metric $\Delta E_P$ was evaluated as the average value on the 200 samples of the test dataset.

The normalized mean squared error for parameter $P$ (with $P = R_{ct}$ or $Q$) can be expressed as follows:

$$NMSE_P = \frac{1}{\mu_{P,estimated} \cdot \mu_{P,nominal}} \cdot \sum_{k=1}^{n} \frac{(P_{estimated,k} - P_{nominal,k})^2}{n} \tag{4}$$

where $\mu_{P,estimated}$ and $\mu_{P,nominal}$ are the average values over the 200 samples of the test dataset for the estimated and nominal value of the parameter $P$, while $n$ is the size of the test dataset (200).

The cost in terms of memory occupation for the ANN trainable parameters was evaluated with the memory usage (*MU*) metric, which defines the number of bytes needed to memorize the ANN trainable parameters, given that each ANN trainable parameter is represented with a floating point number (4 bytes).

### 3.4. Circuit Fitting Accuracy with a PC Software

The 200 samples of the test dataset were used to estimate the simplified Randles circuit parameters using the online EIS data fitting software "EIS Studio" [33]. This test was carried out with all three software-generated datasets, i.e., Dataset A (impedance spectrum using 5 frequencies), Dataset B (impedance spectrum using 15 frequencies), and Dataset C (impedance spectrum using 25 frequencies).

In the case of Dataset A, the EIS data software was not able to fit the impedance spectrum due to the low number of test frequencies. In the case of Dataset B, data fitting was carried out correctly and resulted in $\Delta E_{Rct} = 0.015\%$, $\Delta E_Q = 0.044\%$, $NMSE_{Rct} = 5.98 \cdot 10^{-8}$, and $NMSE_Q = 3.02 \cdot 10^{-7}$. In the case of Dataset C, data fitting was carried out correctly and resulted in $\Delta E_{Rct} = 0.011\%$, $\Delta E_Q = 0.031\%$, $NMSE_{Rct} = 2.33 \cdot 10^{-8}$, and $NMSE_Q = 1.39 \cdot 10^{-7}$.

## 4. Results and Discussion

In this Section, the simulation results for all the ANN structures and the datasets presented in Section 3 are discussed. In particular, in Section 4.1, the performance metrics for the case of the software-generated EIS datasets and all the ANN structures are presented. In Section 4.2, a discussion on the microcontroller SRAM size needed for the implementation of the proposed ANN structures is presented. In Section 4.3, the use of the proposed ANN structures to estimate the EIS parameters for the case of a real dataset in [62] is discussed.

### 4.1. Performance Metrics for the Software-Generated Datasets

The performance metrics for the software-generated datasets are presented in Table 4 for the case of Dataset A, in Table 5 for the case of Dataset B, and in Table 6 for the case of Dataset C. The ANN structures are defined as follows: A (one hidden layer with 8 neurons), B (one hidden layer with 16 neurons), C (two hidden layers with 16 and 8 neurons), D (one hidden layer with 24 neurons), E (two hidden layers with 24 and 12 neurons), F (one hidden layer with 32 neurons), G (two hidden layers with 32 and 16 neurons), H (one hidden layer with 48 neurons), I (two hidden layers with 48 and 24 neurons), J (one hidden layer with 64 neurons), and K (two hidden layers with 64 and 32 neurons).

**Table 4.** Metrics for estimation accuracy and memory usage for different ANN structures with Dataset A.

| ANN Structure | $\Delta E_{Rct}$ | $\Delta E_Q$ | $NMSE_{Rct}$ | $NMSE_Q$ | *MU* (Bytes) |
|---|---|---|---|---|---|
| A | 1.82% | 18.04% | $3.07 \times 10^{-4}$ | $2.66 \times 10^{-2}$ | 424 |
| B | 2.24% | 14.31% | $3.97 \times 10^{-4}$ | $1.87 \times 10^{-2}$ | 840 |
| C | 1.97% | 8.83% | $4.24 \times 10^{-4}$ | $8.44 \times 10^{-3}$ | 1320 |
| D | 1.44% | 13.2% | $1.97 \times 10^{-4}$ | $1.58 \times 10^{-2}$ | 1256 |
| E | 1.73% | 7.37% | $3.26 \times 10^{-4}$ | $6.11 \times 10^{-3}$ | 2360 |
| F | 1.54% | 9.76% | $1.90 \times 10^{-4}$ | $1.03 \times 10^{-2}$ | 1672 |
| G | 1.29% | 5.38% | $1.50 \times 10^{-4}$ | $2.74 \times 10^{-3}$ | 3656 |
| H | 1.03% | 8.15% | $7.37 \times 10^{-5}$ | $7.98 \times 10^{-3}$ | 2504 |
| I | 1.15% | 4.71% | $1.04 \times 10^{-4}$ | $2.24 \times 10^{-3}$ | 7016 |
| J | 1.11% | 9.15% | $1.01 \times 10^{-4}$ | $8.69 \times 10^{-3}$ | 3336 |
| K | 1.43% | 3.04% | $1.99 \times 10^{-4}$ | $1.05 \times 10^{-3}$ | 11,400 |

**Table 5.** Metrics for estimation accuracy and memory usage for different ANN structures with Dataset B.

| ANN Structure | $\Delta E_{Rct}$ | $\Delta E_Q$ | $NMSE_{Rct}$ | $NMSE_Q$ | *MU* (Bytes) |
|---|---|---|---|---|---|
| A | 6.55% | 13.56% | $2.58 \times 10^{-3}$ | $2.19 \times 10^{-2}$ | 1064 |
| B | 2.93% | 10.85% | $7.49 \times 10^{-4}$ | $1.22 \times 10^{-2}$ | 2120 |
| C | 1.82% | 7.57% | $3.40 \times 10^{-4}$ | $5.13 \times 10^{-3}$ | 2600 |
| D | 1.41% | 8.53% | $1.62 \times 10^{-4}$ | $6.72 \times 10^{-3}$ | 3176 |
| E | 2.21% | 5.94% | $4.71 \times 10^{-4}$ | $3.52 \times 10^{-3}$ | 4280 |
| F | 2.01% | 6.54% | $2.97 \times 10^{-4}$ | $4.72 \times 10^{-3}$ | 4232 |
| G | 1.04% | 4.72% | $7.96 \times 10^{-5}$ | $3.02 \times 10^{-3}$ | 6216 |
| H | 0.77% | 5.98% | $6.28 \times 10^{-5}$ | $4.19 \times 10^{-3}$ | 6344 |
| I | 0.90% | 3.44% | $7.28 \times 10^{-5}$ | $1.08 \times 10^{-3}$ | 10,856 |
| J | 0.86% | 4.68% | $6.61 \times 10^{-5}$ | $2.48 \times 10^{-3}$ | 8456 |
| K | 0.95% | 2.64% | $8.55 \times 10^{-5}$ | $7.45 \times 10^{-4}$ | 16,520 |

**Table 6.** Metrics for estimation accuracy and memory usage for different ANN structures with Dataset C.

| ANN Structure | $\Delta E_{Rct}$ | $\Delta E_Q$ | $NMSE_{Rct}$ | $NMSE_Q$ | $MU$ (Bytes) |
|---|---|---|---|---|---|
| A | 4.04% | 14.10% | $1.47 \times 10^{-3}$ | $1.85 \times 10^{-2}$ | 1704 |
| B | 3.66% | 11.64% | $5.99 \times 10^{-4}$ | $1.34 \times 10^{-2}$ | 3400 |
| C | 2.09% | 8.49% | $2.88 \times 10^{-4}$ | $6.91 \times 10^{-3}$ | 3880 |
| D | 1.43% | 10.39% | $1.75 \times 10^{-4}$ | $9.70 \times 10^{-3}$ | 5096 |
| E | 2.88% | 7.87% | $5.56 \times 10^{-4}$ | $5.15 \times 10^{-3}$ | 6200 |
| F | 1.39% | 9.87% | $1.33 \times 10^{-4}$ | $1.03 \times 10^{-2}$ | 6792 |
| G | 1.62% | 6.64% | $2.44 \times 10^{-4}$ | $3.30 \times 10^{-3}$ | 8776 |
| H | 1.19% | 8.53% | $1.24 \times 10^{-4}$ | $7.97 \times 10^{-3}$ | 10,184 |
| I | 1.41% | 4.55% | $1.69 \times 10^{-4}$ | $2.40 \times 10^{-3}$ | 14,696 |
| J | 1.11% | 8.19% | $8.95 \times 10^{-5}$ | $6.23 \times 10^{-3}$ | 13,576 |
| K | 1.25% | 5.01% | $1.19 \times 10^{-4}$ | $1.98 \times 10^{-3}$ | 21,640 |

In Figure 3 (Figure 4), the relative error of the estimated parameter $R_{ct}$ ($Q$) is compared for the different ANN structures and the three considered datasets. As can be seen, the accuracy achieved is not comparable with a PC-based circuit fitting software that, as shown in Section 3.4, is characterized by a relative error lower than 0.05%.

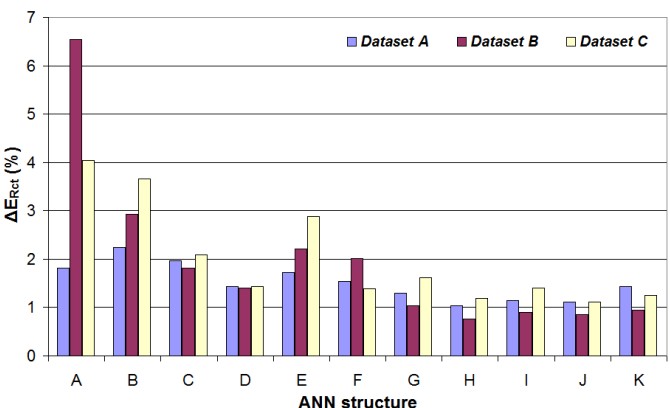

**Figure 3.** Relative error of the estimated parameter $R_{ct}$ for the different ANN structures and the three considered datasets.

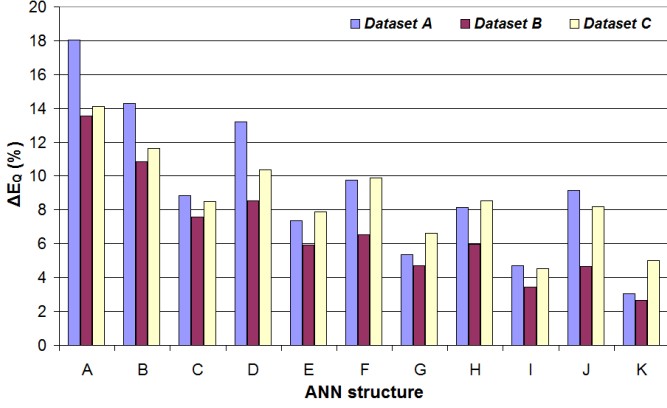

**Figure 4.** Relative error of the estimated parameter $Q$ for the different ANN structures and the three considered datasets.

However, in the case of parameter $R_{ct}$ estimation, a relative error between 1% and 2% can also be achieved for simple ANN structures and Dataset A (with advantages in terms of lower memory occupation), and a relative error lower than 1% can be achieved for more complex ANN structures and Dataset B. In the case of parameter $Q$ estimation, the

situation is worse, and a relative error between 2% and 3% can be achieved only for more complex ANN structures.

In particular, Figure 5 reports the results for Dataset A and ANN structure A. Figure 6 reports the results for Dataset A and ANN structure K. Figure 7 reports the results for Dataset B and ANN structure A. Figure 8 reports the results for Dataset B and ANN structure K. In more detail, Figures 5–8 report the scatter plot showing the estimated $R_{ct}$ value vs. the nominal $R_{ct}$ value for all the 200 samples of the test dataset (a), the scatter plot showing the estimated $Q$ value vs. the nominal $Q$ value for all the 200 samples of the test dataset (b), the Bode plot showing the real component of the impedance (Re(Z)) as function of the test signal frequency (c), and the Bode plot showing the imaginary component of the impedance (Im(Z)) as function of the test signal frequency (d).

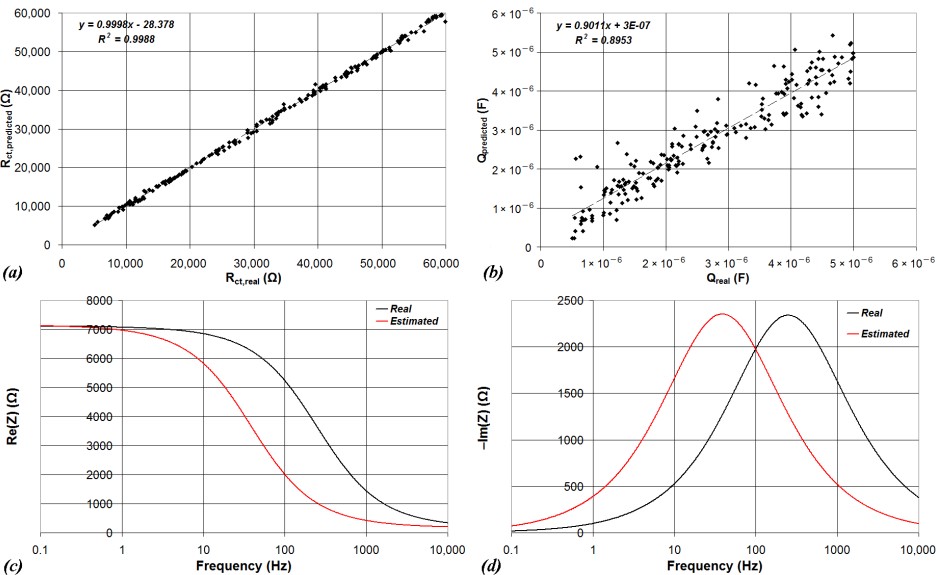

**Figure 5.** Graphical representation of estimated $R_{ct}$ value vs. the nominal $R_{ct}$ value (**a**), estimated $Q$ value vs. the nominal $Q$ value (**b**), Re(Z) vs. frequency (**c**); and −Im(Z) vs. frequency (**d**), in the case of Dataset A and ANN structure A.

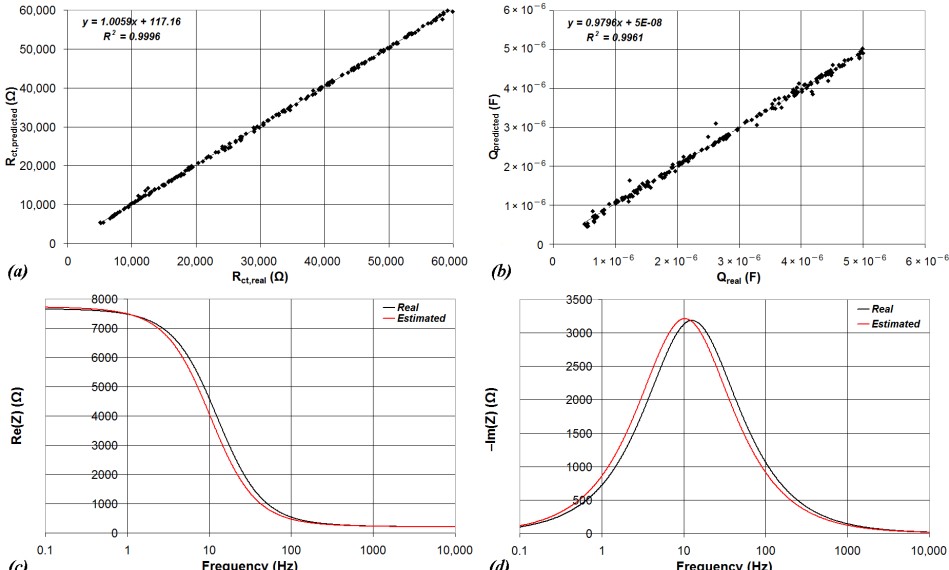

**Figure 6.** Graphical representation of estimated $R_{ct}$ value vs. the nominal $R_{ct}$ value (**a**), estimated $Q$ value vs. the nominal $Q$ value (**b**), Re(Z) vs. frequency (**c**); and −Im(Z) vs. frequency (**d**), in the case of Dataset A and ANN structure K.

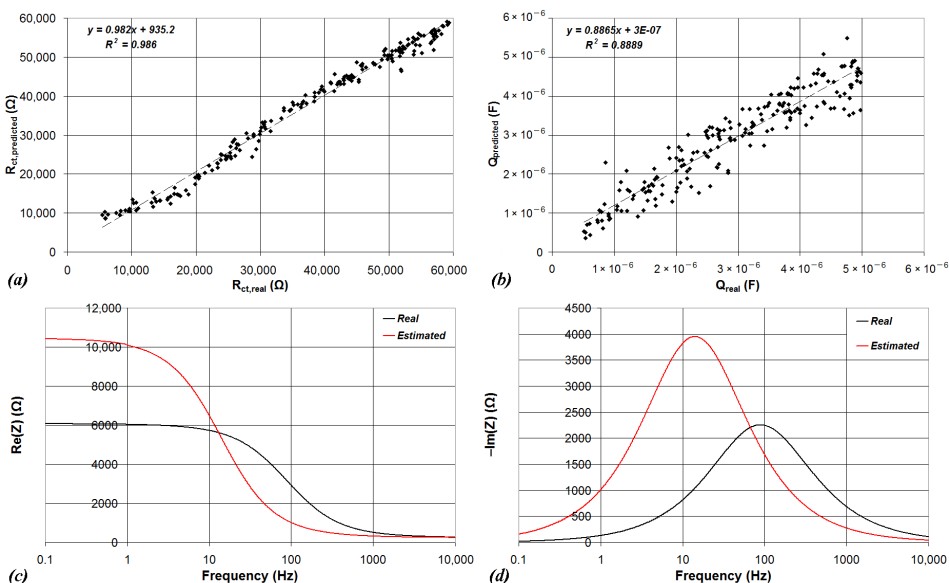

**Figure 7.** Graphical representation of estimated $R_{ct}$ value vs. the nominal $R_{ct}$ value (**a**), estimated $Q$ value vs. the nominal $Q$ value (**b**), Re(Z) vs. frequency (**c**); and −Im(Z) vs. frequency (**d**), in the case of Dataset B and ANN structure A.

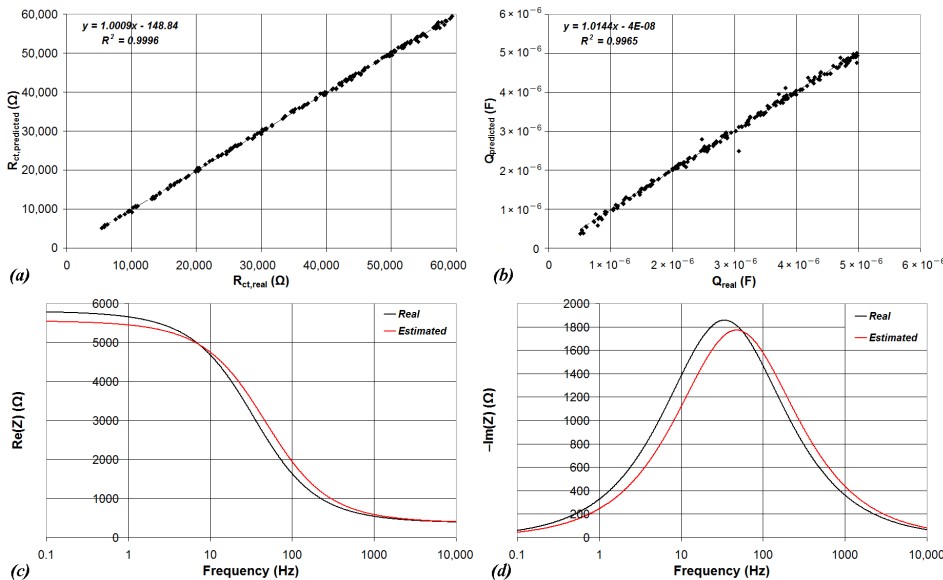

**Figure 8.** Graphical representation of estimated $R_{ct}$ value vs. the nominal $R_{ct}$ value (**a**), estimated $Q$ value vs. the nominal $Q$ value (**b**), Re(Z) vs. frequency (**c**); and −Im(Z) vs. frequency (**d**), in the case of Dataset B and ANN structure K.

The results in the figures are consistent with the data presented in Figures 3 and 4, and show that the charge transfer resistance $R_{ct}$ can be accurately estimated also with an implementation requiring low memory usage (Dataset A, ANN structure A). Instead, the results in the figures show that the accurate estimation of the parameter $Q$ requires a more complex ANN structure.

In Figure 9 (Figure 10), the relative error of the estimated parameter $R_{ct}$ ($Q$) is plotted versus the memory usage (*MU*). In the case of the parameter $R_{ct}$ estimation, the best accuracy (relative error of 0.77%) is achieved for ANN structure H and Dataset B with a relatively low *MU* of 6344 bytes, while a relative error close to 1% (1.11%) can be achieved with ANN structure J and Dataset A with a *MU* of only 3336 bytes. For very low-cost microcontrollers that feature a low SRAM size, the only option is to use ANN structure

A with Dataset A, with a *MU* of only 424 bytes and a relative error of 1.82%. In the case of parameter *Q* estimation, to achieve an acceptable accuracy, a *MU* higher than 10 kB is needed, with the best results that are a relative error of 3.04% for ANN structure K and Dataset A (*MU* of 11,400 bytes), and a relative error of 2.64% for ANN structure K and Dataset B (*MU* of 16,520 bytes).

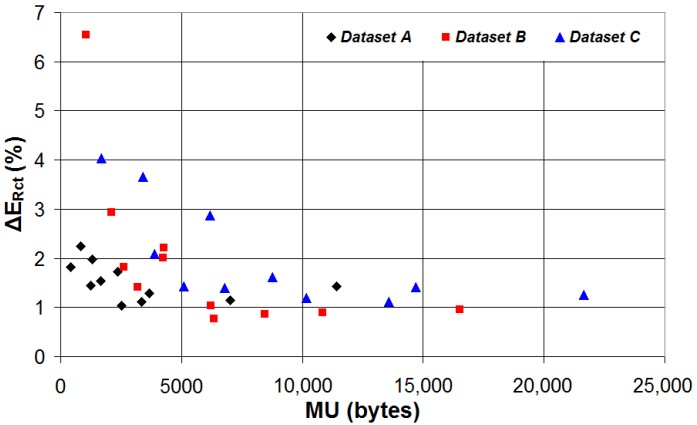

**Figure 9.** Relative error of the estimated parameter $R_{ct}$ plotted versus the memory usage (*MU*).

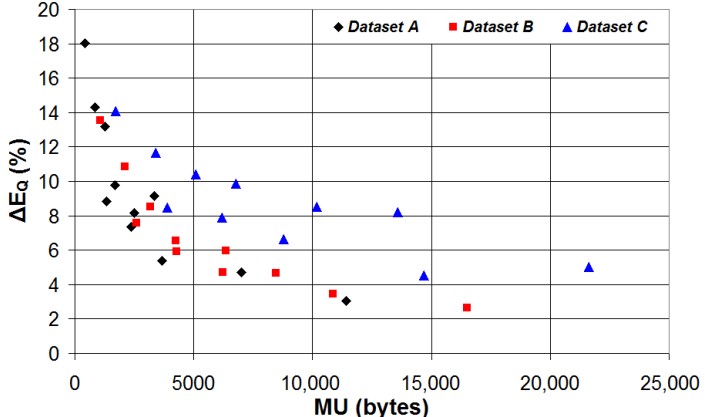

**Figure 10.** Relative error of the estimated parameter *Q* plotted versus the memory usage (*MU*).

### 4.2. Discussion on the Required SRAM Size

In the previous section, the size of the SRAM required to store the ANN trainable parameters has been estimated for all considered ANNs, and for the three considered datasets (Tables 4–6). The size of the SRAM required for the implementation of an EIS-based biosensor measurement system and equivalent circuit parameters estimation can be evaluated as the sum of the following four components:

1.  The memory needed to store the voltage sine-wave signals $V_{in}$ (input test signal) and $V_{out}$ (proportional to the current through the sensor), acquired with an ADC (either integrated in the microcontroller or external) and used to calculate the sensor impedance components Re(Z) and Im(Z). The signals $V_{in}$ and $V_{out}$ must be acquired for every test frequency. However, since the impedance components are calculated immediately after the signals' acquisition, the same memory region can be reused for the signals' acquisition for different test frequencies. Assuming that each sample is stored as a floating point number (4 bytes) and 100 samples are acquired for each of the two signals ($V_{in}$ and $V_{out}$), 800 bytes are needed.

2.  The memory needed to store the sensor impedance components Re(Z) and Im(Z) for each frequency of the test signal. Assuming that each impedance component is

represented by a floating-point number (4 bytes), this memory component requires 40 bytes for Dataset A (5 test frequencies), 120 bytes for Dataset B (15 test frequencies), and 200 bytes for Dataset C (25 test frequencies).

3. The memory needed to store the trainable parameters of the ANNs. The size of this memory component is presented in Tables 4–6 and ranges from 424 bytes to 11,400 bytes for Dataset A, from 1064 bytes to 16,520 bytes for Dataset B, and from 1704 bytes to 21,640 bytes for Dataset C.

4. The memory needed to execute the microcontroller code: acquisition of the sine-wave signals, calculation of the impedance components, implementation of the ANN sum of products and application of the activation function, information data transfer with the UART interface. This memory component was estimated by the implementation on a Nucleo-L152RE development board of the code written in C and compiled with the MBED Keil Studio Cloud online compiler. The SRAM size was estimated to be about 2 kB, but it can eventually be lowered by a more efficient assembly code.

By summing the size of the different memory components, the microcontroller SRAM size requirements can be estimated in the range from 3312 bytes to 14,288 bytes for Dataset A, from 4032 bytes to 19,488 bytes for Dataset B, and from 4752 bytes to 24,688 bytes for Dataset C.

Different prototypes of portable and wearable sensor systems have been discussed in the literature for different types of applications. A low-cost, portable monitoring system for indoor environment quality, based on the ATSAMD21G18 microcontroller with 32 kB of SRAM, was presented in [4]. A portable battery-operated sensor system for the measurement of peroxide index and total phenolic content in olive oil, based on the STM32L152VCT6 microcontroller with 32 kB of SRAM, was presented in [6]. A portable sensor system, based on the STM32L152RE microcontroller with 80 kB of SRAM, that exploits optical attenuation measurements for the evaluation of the solid fat content in vegetable fats and oils was presented in [8]. A wireless-enabled, portable, potentiometric biosensor for bacterial concentration detection in urine, based on the Tensilica LX6 microcontroller with 520 kB of SRAM, was presented in [19]. An electrochemical sensor, based on the AT91SAM3X8E microcontroller with 96 kB of SRAM, that features a gas sensor array for the determination of fish quality was presented in [66]. A portable electrochemical sensing platform, based on the STM32F303RET6 microcontroller with 80 kB of SRAM, that exploits EIS measurements for the measurement of atrazine concentration was presented in [67]. A wearable photoplethysmographic (PPG) sensor system, based on the CY8C29466 microcontroller with 2 kB of SRAM, for the measurement of changes in volume of blood vessels was presented in [68]. An IoT wearable device, based on the nRF52840 microcontroller with 256 kB of SRAM, that can be used to track the health and recovery of COVID-19 patients was presented in [69].

Based on the presented literature review, almost all microcontrollers integrated in the proposed systems, with the exception of the system presented in [68], feature a SRAM size higher than 32 kB that allows the implementation of the proposed approach for EIS data fitting for all ANN structures and all the considered datasets.

### 4.3. Validation on a Real EIS Dataset

In this section, the proposed approach for EIS data fitting using simple ANN structures is validated on the real dataset presented in [62]. The dataset has been obtained by experimental measurements using an electrochemical biosensor functionalized with concanavalin A on gold microelectrodes to detect dengue virus in biological samples. As discussed in Section 3.1, it is composed of 18 human serum samples: 6 samples are positive

for dengue fever (DF), 6 samples are positive for dengue hemorrhagic fever (DHF), and 6 samples are negative for dengue virus (DN, i.e., dengue negative).

The real dataset presented in [62] has been tested with three different ANN structures: ANN structure A (one hidden layer with 8 neurons) with the impedance characteristic samples on five test frequencies from 0.1 Hz to 10 kHz (hereafter referred to as ANN_A_5, with a memory usage of 3312 bytes when implemented on a microcontroller); ANN structure J (one hidden layer with 64 neurons) with the impedance characteristic samples on five test frequencies from 0.1 Hz to 10 kHz (hereafter referred to as ANN_J_5, with a memory usage of 6224 bytes when implemented on a microcontroller); ANN structure K (two hidden layers with 64 and 32 neurons) with the impedance characteristic samples on 15 test frequencies from 0.1 Hz to 10 kHz (hereafter referred to as ANN_K_15, with a memory usage of 19,488 bytes when implemented on a microcontroller).

The results are presented in Figure 11, where the values of the charge transfer resistance $R_{ct}$ and the capacitance $Q$ are plotted for the 18 tested samples, for the case of parameters estimated with ANN_A_5 (a), ANN_J_5 (b), ANN_K_15 (c), and for the case of the nominal values of $R_{ct}$ and $Q$ presented in [62] (d). As can be seen, all the tested ANN structures can accurately discriminate between samples contaminated by dengue virus (DF or HDF) and samples that are not contaminated by dengue virus (DN). Regarding the discrimination between DF and HDF samples, only the ANN structures ANN_J_5 and ANN_K_15, as well as the reference parameter values presented in [62], can differentiate the two subgroups of dengue fever.

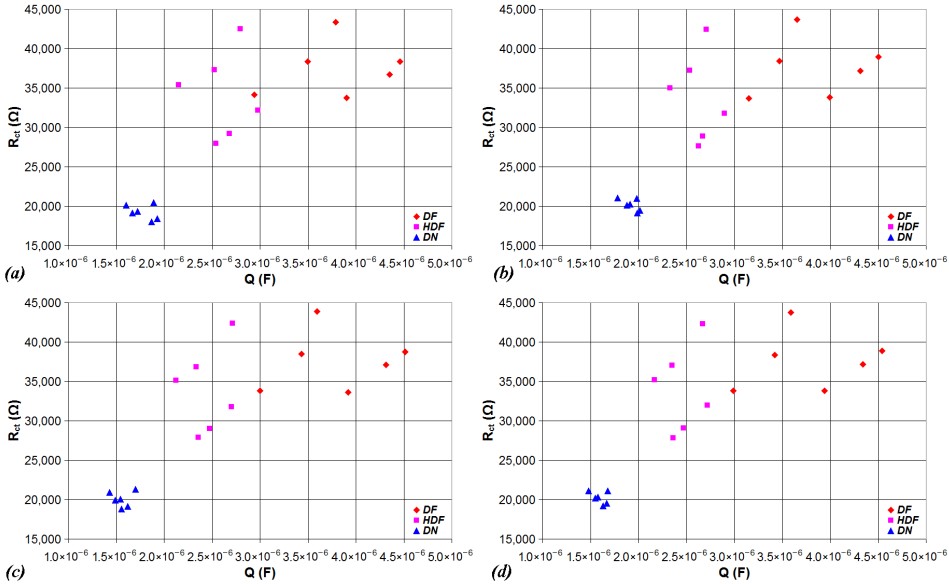

**Figure 11.** Scatter plots of the charge transfer resistance $R_{ct}$ vs. the capacitance $Q$ for the 18 tested samples in the case of parameters estimated with ANN_A_5 (**a**), ANN_J_5 (**b**), ANN_K_15 (**c**); and in the case of the nominal values of $R_{ct}$ and $Q$ presented in [62] (**d**).

## 5. Conclusions

This work presents a study on the feasibility of using simple structures of Artificial Neural Networks (ANNs) in biosensor data analysis for implementation on resource-constrained microcontrollers. We have considered the case study of biosensors based on Electrical Impedance Spectroscopy (EIS) that are modeled using the simplified Randles circuit. We have investigated different ANN structures of maximum two hidden layers with three different software-generated datasets, as well as with a real dataset from the literature, sampling the impedance spectrum between 0.1 Hz and 10 kHz with 5, 15, and 25 test frequencies, respectively.

The results have shown that the achieved accuracy in the equivalent circuit parameters estimation is lower than the accuracy achieved using a PC software for EIS data fitting, but a relative error in the order of 1% can be achieved for the estimation of the charge transfer resistance, with a memory usage (to store the ANN trainable parameters) of about 3 kB. This value is compatible with the implementation on low-cost microcontrollers with limited SRAM size. In the case of the biosensor equivalent capacitance estimation, the minimum relative error is higher (about 3%) with a memory usage in the order of 10–15 kB.

**Author Contributions:** Conceptualization, M.G.; methodology, M.G.; software, M.G.; validation, M.G.; formal analysis, M.G.; investigation, M.G.; resources, M.G.; data curation, M.G.; writing—original draft preparation, M.G.; writing—review and editing, M.G. and M.O.; visualization, M.G. and M.O.; supervision, M.G. and M.O.; project administration, M.G. and M.O. All authors have read and agreed to the published version of the manuscript.

**Funding:** This research received no external funding.

**Data Availability Statement:** The original contributions presented in this study are included in the article. Further inquiries can be directed to the corresponding author.

**Conflicts of Interest:** The authors declare no conflicts of interest.

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
