# Peer review of "Data Analysis of Electrical Impedance Spectroscopy-Based Biosensors Using Artificial Neural Networks for Resource Constrained Devices"

_jlpea, doi:10.3390/jlpea15040056_

Round 1

Reviewer 1 Report

Comments and Suggestions for Authors

What is the sum of the data capacity required for impedance measurement based on EIS, the data capacity to store them, and the capacity for data analysis (10 kB to 15 kB)? What is the memory range of microcomputers for portable and wearable sensors? If we show whether the memory is larger than the order of 10kB to 15kB obtained in Figures 5 and 6, readers will understand whether the data analysis using the neural network is satisfactory. For this purpose, in Fig. 3, please compare the impedance curve (actual data) and the fitted curve when using data set B and ANN structures A and K. Show how different the fitted curves are when ΔE_Rct = 6.5% and less than 1%. The same applies to Fig. 4 (ΔE_Q). Also in Figs. 5 and 6, compare the fitted curves when ΔR_Rct is particularly large and when it is the smallest: (i) ANN structure A and dataset A@MU = 3336 bytes; (ii) ANN structure K and dataset A@MU=11400 bytes; and (iii) ANN structure K and dataset B@MU= 16520 bytes).

Another points are:

  • Line 68 (Table 1): What is SRAM, ADC, DAC?
  • line 176: What is "ReLu"?

Reviewer 2 Report

Comments and Suggestions for Authors

< !--StartFragment -->

The authors explore the use of simple Artificial Neural Networks (ANNs) for fitting electrochemical impedance spectroscopy (EIS) data, which can run on microcontrollers with limited memory. The findings showed that simple ANNs provided a low-power, low-cost solution for fitting a simplified Randles circuit, with accuracy between 1% and 3%.

The manuscript is well written and structured, but some improvements are recommended before it can be accepted.

  1. The phrase “Electrochemical biosensors are devices that use it (EIS) to measure a target analyte”, in line 13, is not entirely accurate, as electrochemical biosensors can also be based on other electrochemical techniques, such as square wave voltammetry, differential pulse voltammetry, cyclic voltammetry, and chronoamperometry. Electrochemical sensors specifically investigated using the EIS technique are known as impedimetric sensors.
  2. In the sentence "Electrochemical biosensors are widely used for the detection of different types of analytes" (line 95), I suggest including reference 10.1016/S0956-5663(01)00115-4. This will also help avoid limiting electrochemical biosensors to just the EIS technique. Additionally, the authors described the main components of an electrochemical cell used for sensing. I missed references that discuss this topic, regardless of the size and architecture of the systems, such as the papers 10.1021/acsmeasuresciau.2c00070, 10.1039/D0TA05796G, and 10.1039/CS9942300289.
  3. Figures 3, 4, 5, and 6 can be used as panels in a single figure. Additionally, the points in Figures 5 and 6 can be colored as in Figures 3 and 4 to specify the analyzed data set, whether they are from Dataset A, B, or C.
  4. I suggest that the authors enrich the introduction by citing some works that already employ ANN, DNN, and machine learning to adjust EIS spectra, even if they are not in the field of sensors/biosensors: 10.3390/en14092526, 10.1002/aisy.202300085, 10.1016/j.electacta.2025.146231, 10.1016/j.electacta.2025.146231.
  5. As the manuscript focuses more on feasibility than on a breakthrough in either ANN theory or biosensor technology, it would be interesting if the authors could add at least one real EIS dataset to demonstrate the operation of the ANN trained with the dataset generated with LabVIEW.

Round 2

Reviewer 2 Report

Comments and Suggestions for Authors

The authors have addressed all concerns and comments appropriately, and I support the publication of the manuscript in its current form.